# The Synthesis, Characterization, and Fluxional Behavior of a Hydridorhodatetraborane

**DOI:** 10.3390/molecules28186462

**Published:** 2023-09-06

**Authors:** Fatou Diaw-Ndiaye, Pablo J. Sanz Miguel, Ricardo Rodríguez, Ramón Macías

**Affiliations:** Departamento de Química Inorgánica, Instituto de Síntesis Química y Catálisis Homogénea (ISQCH), Universidad de Zaragoza-CSIC, 50009 Zaragoza, Spain

**Keywords:** boranes, metallaboranes, metal hydrides, fluxionality, solid-state structure

## Abstract

The octahydridotriborate anion plays a crucial role in the field of polyhedral boron chemistry, facilitating the synthesis of higher boranes and the preparation of diverse transition metal complexes. Among the stable forms of this anion, CsB_3_H_8_ (or (n-C_4_H_9_)_4_N)[B_3_H_8_] have been identified. These salts serve as valuable precursors for the synthesis of metallaboranes, wherein the triborate anion acts as a ligand coordinating to the metal center. In this study, we have successfully synthesized a novel rhodatetraborane dihydride, [Rh(*η*^2^-B_3_H_8_)(H)_2_(PPh_3_)_2_] (**1**), which represents a Rh(III) complex featuring a bidentate chelate ligand fasormed by B_3_H_8_^−^. Extensive characterization of this rhodatetraborane complex has been performed using NMR spectroscopy in solution and X-ray diffraction analysis in the solid state. Notably, the complex exhibits intriguing fluxional behavior, which has been investigated using NMR techniques. Moreover, we have explored the reactivity of complex **1** towards pyridine (py) and dimethylphenylphosphine (PMe_2_Ph). Our findings highlight the labile nature of this four-vertex rhodatetraborane as it undergoes disassembly upon attack from the corresponding Lewis base, resulting in the formation of borane adducts, LBH_3_, where L = py, PMe_2_Ph. Furthermore, in these reactions, we report the characterization of new cationic hydride complexes, such as [Rh(H)_2_(PPh_3_)_2_ (py)]^+^ (**2**) and [Rh(H)_2_(PMe_2_Ph)_4_]^+^. Notably, the latter complex has been characterized as the octahydridotriborate salt [Rh(H)_2_(PMe_2_Ph)_4_][B_3_H_8_] (**3**), which extends the scope of rhodatetraborane derivatives.

## 1. Introduction

The synthesis of [*arachno*-B_3_H_8_]^−^ has gathered significant attention in recent years due to its pivotal role in the preparation of higher boranes, such as *closo*-B_12_H_12_^2−^, and a wide array of transition metal complexes [1,2,3,4].

Many salts of this anion have been prepared using B_2_H_6_, but concerns regarding the toxicity and flammability of diborane have prompted the exploration of alternative synthetic routes. Among the various methods, the reaction of Na[BH_4_] with I_2_ stands out [2]; however, the presence of iodide anions poses challenges in obtaining the desired product. Recent investigations have focused on replacing iodine with various metal halides as oxidants, offering improved efficiency and selectivity in the synthesis of [*arachno*-B_3_H_8_]^−^.

The crystal structure of [(H_3_N)_2_BH_2_][B_3_H_8_] has been determined by Peters and Nordman, shedding light on the structural composition of the octahydridotriborate anion. The anion consists of a triangular arrangement of boron atoms, with two bridging and six terminal hydrogens [3].

The inherent stability and accessibility of the *arachno*-B_3_H_8_^−^ anion have paved the way for extensive investigations into its reactivity with various metal complexes [4]. The resulting *arachno*-2-metallaboranes can be synthesized through ligand substitution reactions, offering a general route for the incorporation of transition metals into the cluster framework:

L_n_MX + B_3_H_8_^−^
→ [L_n−1_MB_3_H_8_]
+ L + X^−^ (L = neutral ligand; X = halide)



The reactivity of *arachno*-B_3_H_8_^−^ has been explored with a wide range of transition metals, spanning across the periodic table, including Ti, Cr, Mo, W, Mn, Re, Fe, Ru, Os, Ir [5], Cu, Ag and Zn [1,3,4].

The reaction between a transition metal complex and the octahydridotriborate anion is, *a priori*, the most direct route to the synthesis of *arachno*-2-metallatetraboranes. However, some of the reported metallatetraboranes were prepared from reactions of either monocyclopentadienyl metal chlorides or hydride-ligated complexes of transition metals from groups 5–9 with monoboranes (LiBH_4_ or BH_3_THF) [6,7,8]. This synthetic procedure was developed mainly by Fehlener and co-workers at Notre Dame University (Notre Dam, IN, USA) [9]; and Gosh and co-workers have been using it for a good number of years, at the Indian Institute of Technology Madras (Chennai, India), in the pursuit of new metallaboranes [10].

Alternatively, reactions between metal complexes and larger boranes, such as pentaborane, were also a route to tetraboranes, via cluster dismantling processes [11].

To expand the scope of transition element *arachno*-metallaboranes and explore novel structures and dynamic processes, our study focused on investigating the reactivity between Wilkinson’s catalyst, [RhCl(PPh_3_)_3_], and the octahydridotriborate anion, [B_3_H_8_]^−^. This investigation resulted in the successful synthesis of dihydridorhodatetraborane, [Rh(η^2^-B_3_H_8_)(H)_2_(PPh_3_)_2_] (**1**). The compound was comprehensively characterized using NMR spectroscopy and X-ray diffraction analysis. Notably, the newly synthesized metallatetraborane exhibited a chemical non-rigidity, which was studied using NMR spectroscopy at variable temperatures. In addition, we have carried out an exploratory study of the reactivity of **1** with Lewis bases, resulting in the characterization of hydride metal complexes.

## 2. Results and Discussion

### 2.1. Synthesis of [Rh(η^2^-B_3_H_8_)(H)_2_(PPh_3_)_2_] (***1***)

The reaction of the Wilkinson’s catalyst with the cesium salt CsB_3_H_8_ in ethanol leads to the formation of dihydridorhodatetraborane (**1**) (Figure 1).

Due to the limited solubility of the cesium salt in ethanol and the insolubility of the rhodium complex (Wilkinson’s catalyst) in the same solvent, the reaction proceeds in a heterogeneous solid–liquid phase. The reaction mixture initially forms a brick-red suspension, which transforms into a red-orange product, corresponding to the formation of dihydridorhodatetraborane (**1**). The product is then collected by filtration using a sintered disc filter funnel.

As an alternative approach, we conducted the reaction using the *tris*(dioxane) solvate NaB_3_H_8_^.^3(C_4_H_8_O_2_) as the starting material in diethyl ether, which also exhibits limited solubility for Wilkinson’s catalyst. Similar to the ethanol system, this synthesis is characterized by a heterogeneous reaction. The resulting yellow product, identified as hydridorhodathiaborane, is easily filtered under ambient conditions, yielding **1**.

### 2.2. X-ray Diffraction Analysis

The Cambridge Crystallographic Data Centre (CCDC) provides X-ray diffraction analyses for fifteen *arachno*-metallaboranes, incorporating {ML_n_}-fragments of Nb [7], Cr [12], Mo [10], W [6,10], Mn [13], Re [8,13], Ru [11,14,15], Os [16] and Cu [17]. However, the availability of comparative structural data across the periodic table, for this particular class of four-vertex *arachno*-metallaboranes, is limited. It is important to emphasize that the crystal structure of compound **1** represents the first example of a Group 9 *arachno*-2-metallaborane characterized by X-ray diffraction analysis.

Single crystals of the compound were obtained by diffusing hexane into a solution of **1** in CH_2_Cl_2_. Figure 1 depicts an ORTEP-type drawing, illustrating selected interatomic distances and angles. The rhodium center in the compound exhibits an octahedral coordination sphere, where the B_3_H_8_^−^ moiety acts as a bidentate *η*^2^-ligand through two B–H–Rh bridge bonds. These bridge bonds are located *trans* to the *exo*-polyhedral hydride ligands. Completing the coordination number 6 around the metal, two Ph_3_P ligands are mutually *trans*. Consequently, the molecule can be classified as an eighteen-electron, six-coordinate, octahedral *d*^6^ rhodium(III) complex.

Alternatively, compound **1** can be described as a four-vertex *arachno*-cluster, which can be related to the parent *arachno*-B_4_H_10_ by replacing a BH_2_ ‘wing-tip’ with the *d*^6^-{Rh(H)_2_(PPh_3_)_2_} fragment. This description follows the architectural patterns proposed by Williams [18], where the four-vertex butterfly type *arachno*-cluster is derived from an octahedron by removing two adjacent vertices. According to the polyhedral skeletal electron pair theory (PSEPT) [19,20], these clusters are expected to have seven skeletal electron pairs (n + 3; where n is the number of vertices of the polyhedral cluster). Applying the PSEPT electron-counting rules, the {Rh(H)_2_(PPh_3_)_2_} group in compound **1** can be considered as a vertex contributing three electrons to the cluster framework bonding [9 e^−^(Rh) + 4 e^−^(2PPh_3_) + 2 e^−^(2H) − 12 e^−^ = 3 e^−^], resembling the BH_2_ ‘wing-tip’ in *arachno*-B_4_H_10_.

The distances between Rh2 and P1, as well as Rh2 and P2, are determined to be 2.3103(11) Å and 2.2911(11) Å, respectively. These bond lengths are significantly shorter compared to the Ru–P lengths observed in the *arachno*-2-ruthenatetraborane, [Ru(*η*^2^-B_3_H_8_)(H)_2_(PPh_3_)_2_], which also features two mutually *trans* PPh_3_ ligands. In the ruthenatetraborane, the Ru–P bond lengths are measured to be 2.373(1) Å and 2.364(1) Å. However, when the hydrotris(pyrazol-1-yl)borate-ligated ruthenatetraborane, [Ru(*η*^2^-B_3_H_8_)(PPh_3_){*K*^3^-HB(pz)_3_}], is considered, the Ru–P bond distance is slightly shorter at 2.317(1) Å. Analysis from the Cambridge Crystallographic Data Centre (CCDC) reveals that the mean bond length for Rh–PPh_3_ is 2.318 Å, while the mean bond distance for Ru–PPh_3_ is slightly longer at 2.350 Å. Based on these findings, it can be concluded that the M–P bond distances fall within the crystallographic data, indicating that the Ru–PPh_3_ bonds are on average longer than the corresponding Rh–PPh_3_ lengths.

It is noteworthy to highlight that the {Ru(CO)(H)(PPh_3_)_2_} and {Ru(PPh_3_){*K*^3^-HB(pz)_3_} fragments present in the ruthenaboranes are isolobal and isoelectronic with the {Rh(H)_2_(PPh_3_)_2_} group observed in compound **1**. These fragments contribute three electrons to the cluster framework, thus fulfilling the expected seven skeletal electron pairs (seps) characteristic of a four-vertex *arachno*-cluster.

In the crystal structure of compound **1**, the rhodatetraborane clusters exhibit a packing arrangement along the crystallographic axis *a*. These clusters form ribbons that associate in pairs through sextuple phenyl embrace (SPE) interactions. The separation between the P atoms (P···P) and the collinearity of Rh–P···P–Rh are measured at 7.9 Å and 165°, respectively, falling within the reported range for this attractive edge-to-face interaction (Figure 2). The SPE interaction arises from intermolecular edge-to-face C–H···π attractive forces facilitated by the presence of phenyl rings [21,22].

All metal octahydridotriboranes stored in the CCDC exhibit a notable similarity, with mean distance values of 1.745 Å, 1.797 Å and 1.794 Å for the B1–B3, B1–B4 and B3–B4 linkages, respectively. Among these linkages, the B1–B3 edge involved in the M–B–B interaction (M = Nb, Cr, Mo, W, Mn, Re, Ru, Os, Cu) shows the shortest mean value. However, the length of the B1–B3 edge varies within the range of 1.707 Å to 1.833 Å, which is larger than the range of 1.772 Å to 1.817 Å observed for the other two B–B connections involving B–B–B bonds. This structural feature is expected due to the variation in the metal center across the metallatetraborane series. The M–B–B interaction, facilitated by two bridging hydrogen atoms, is expected to influence the B1–B3 distance, leading to significant differences among the compounds.

In the parent *arachno*-B_4_H_10_ cluster, the ‘hinge’ B1–B3 edge exhibits a mean value of 1.722 Å, which is the shortest among the five B–B bond distances present in this four-vertex *arachno*-cluster. The other B–B distances in *arachno*-B_4_H_10_ range between 1.844 Å and 1.847 Å, based on the average values derived from the five structures available in the CCDC. The butterfly dihedral angle, characterizing the molecular structure of B_4_H_10_, is measured to be 118.4 ± 0.4°. In comparison, the dihedral angle for the metal compounds falls within the range of 124.5 ± 5.2°. The smallest dihedral angle observed among the metallatetraboranes is 119.3°, which corresponds to the octahydridotriborato-*bis*(triphenylphosphine)copper(I) compound [17]. This compound features a {(PPh_3_)_2_Cu} vertex with both *endo*- and *exo*-triphenyl ligands, closely resembling the {BH_2_} vertex in *arachno*-tetraborane(10). The coordination of the {(PPh_3_)_2_Cu} and {BH_2_} vertices, bound to the {{*η*^2^-B_3_H_8_} fragment, exhibits a tetrahedral geometry. Consequently, significant structural similarities between these two molecules are expected, as evidenced by the similarity in their dihedral angles.

Among the tetrametallaboranes determined through crystallography and deposited in the CCDC, different metal fragments act as vertices, formally replacing the {BH_2_} vertex in *arachno*-B_4_H_10_. As a result, notable differences in the dihedral angles are observed, depending on the nature of the *endo*- and *exo*-ligands. For instance, the [Ru(*η*^2^-B_3_H_8_)(Cl)(*η*^6^-(CH_3_)_6_C_6_))] cluster features a *pseudo*-octahedral ruthenium(II) center, bonded to *exo*-hexamethylbenzene and *endo*-chloro ligands. This metal fragment leads to a relatively flat structure with a dihedral angle of 129.7°. Another tetrametallaborane, [Nb(*η*^2^-B_3_H_8_)(*η*^5^-(C_5_H_5_)_2_], exhibits a dihedral angle of 125.7°. In this case, the coordination number around the niobium(III) center is eight, with six positions occupied by the cyclopentadienyl ligands, each acting as a tridentate ligand, and the remaining two positions occupied by the bidentate octahydridotriborate anion.

### 2.3. NMR Characterization and Comparison

The assignments provided in Table 1 for the resonances are reasonably determined based on their relative intensities and by comparing them with the resonances observed in the previously reported hydridoiridatetraborane, [Ir(*η*^2^-B_3_H_8_)(H)_2_(PPh_3_)_2_] [5]. To further support these assignments, DFT calculations were performed on a model compound, [Rh(*η*^2^-B_3_H_8_)(H)_2_(PH_3_)_2_]. The calculated data confirmed the resonance assignments and provided additional insight into the molecular structure and behavior of the compound.

The NMR data obtained for compound **1** are fully in accord with the solid-state structure determined by X-ray diffraction analysis. The room temperature ^11^B NMR spectrum shows two distinct resonances at δ(^11^B) −1.0 and −38.9 ppm, with a relative intensity ratio of 1:2. These resonances can be attributed to the B4 and B1,3 vertices, respectively. Interestingly, the uncoupled ^11^B spectrum does not display the expected ^1^*J*(^11^B-^1^H) coupling constants (refer to Appendix A). This observation can be rationalized considering the chemical non-rigidity of the *endo*- and bridging-hydrogen atoms present in the {(*η*^2^-B_3_H_8_)} ligand (*vide infra*).

Figure 3 illustrates a stick representation of the chemical shifts and relative intensities in the ^11^B spectra for a series of isostructural and isoelectronic *arachno*-metallatetraboranes similar to compound **1**. These four-vertex clusters exhibit highly similar overall ^11^B shielding patterns. The resonances corresponding to the metal-bound B1–B3 positions are grouped within a narrow region of δ(^11^B) from −36.0 to −41.0 ppm. On the other hand, the signals associated with the “wing-tip” B4 position are observed between δ(^11^B) −1.0 and +6.0 ppm.

The observed marked similarity in the ^11^B NMR resonances is somewhat surprising, considering that the metal fragments change from early to late-transition elements, each bearing different ligands such as CO, PPh_3_, dppe, C_5_H_5_, C_5_Me_5_ and hydrides, and the fact that the spectra were recorded in different solvents such as toluene-d^8^, CD_2_Cl_2_ and CDCl_3_.

This finding suggests that the fundamental nature of the metal-to-{*η*^2^-B_3_H_8_} fragment interaction is maintained throughout this series of compounds. According to the electron-counting rules [19,20] the {Nb(*η*^5^-C_5_H_5_)_2_}- [7], {W(PMe_3_)_3_(H)_3_}- [6], {Re(CO)_4_}- [5], {Mn(CO)_2_(dppe)}- [13], {Os(CO)(PPh_3_)_2_(H)}- [5], {Ru(CO)(PPh_3_)_2_(H)}-, {Fe(*η*^5^-C_5_Me_5_)(CO)}-, {Ir(H)_2_(PPh_3_)_2_} [5] and {Rh(H)_2_(PPh_3_)_2_}-vertices contribute three electrons to the cluster framework. This electron count yields seven seps, as expected for four-vertex arachno-2-metallatetraboranes. The consistency in electron counting and the resultant ^11^B resonances further support the notion that the interaction between the metal fragment and the {*η*^2^-B_3_H_8_} ligand is maintained across this series of compounds.

At 298 K, the ^1^H-{^11^B} NMR spectrum of compound **1** exhibits three signals at δ(^1^H) +2.66 ppm, −6.96 ppm and −11.95 ppm, with a relative intensity ratio of 1:2:2. These spectroscopic data do not match the expected pattern based on the molecular structure of the dihydrorhodatetraborane. According to the C_s_ point group symmetry, we would anticipate five proton resonances with a relative intensity ratio of 1:1:2:2:2, along with aromatic Ph signals (30H). However, when the ^1^H-{^11^B} spectrum is measured at 223 K, the expected pattern is observed. Peaks appear at δ(^1^H) +2.54 (1H), +1.82 (1H), −0.11 (2H), −1.11 (2H), −7.07 (2H) and −11.79 ppm.

The lowest frequency signal at −11.79 ppm, corresponding to the Rh–H hydride ligands, does not broaden in the proton-coupled spectrum. This hydride resonance exhibits the characteristic pattern of a broad quintet, which appears as an apparent broad triplet in the ^1^H-{^31^P} spectrum (refer to Appendix A). However, the chemically equivalent hydride ligands, H2 and H6 in Figure 1, couple unequally to the H1,2 and H2,3 nuclei, resulting in magnetic non-equivalence (Appendix A). Consequently, the ^1^H-{^31^P} spectrum for the Rh–H ligands (H2, H6 in Figure 1) displays second-order behavior (Appendix A).

In the dihydridoiridatetraborane analogue, [Ir(*η*^2^-B_3_H_8_)(H)_2_(PPh_3_)_2_], the hydride signal appears at δ(^1^H) −13.30 p.p.m, appearing as a triplet of doublets due to cisoid coupling to two ^31^P nuclei with very similar coupling constants. Additionally, a small transoid coupling, ^2^J(^1^H_bridge_-^1^H_t_) = 7.0 Hz, is observed. In compound **1**, there are also two ^31^P nuclei with similar coupling constants. However, the proton pattern of the hydride nuclei shows second-order effects, as discussed above. Interestingly, the calculated transoid ^2^J(^1^H1,2-^1^H6) coupling constant in compound **1** is significantly larger compared to that observed in the dihydridoiridatetraborane analogue [5].

The two ^31^P nuclei in compound **1** form an AB-spin system with strong coupling, which is evident from the presence of a “roof effect” in the ^31^P-{^1^H} spectrum at 202 MHz (refer to Appendix A). In a strong coupling regime, the separation of the two central states is determined by the formula C = [((δυ)^2^ + J^2^]^½^, where δυ represents the difference in resonance frequencies of the two spins and J is the scalar coupling constant [23]. The large ^2^J(^31^P1–^103^Rh–^31^P2) of 367 Hz indicates a mutually trans-disposition of the two phosphorus atoms, confirming the molecular structure determined by X-ray diffraction analysis (Figure 1).

The proton signals at δ(^1^H) +2.54 (1H), +1.82 (1H), −0.11 (2H), −1.11 (2H) and −7.07 (2H) exhibit significant broadening in the ^1^H spectrum compared to the ^1^H-{^11^B} spectrum. This indicates that these resonances correspond to ^1^H nuclei directly bound to boron atoms. The molecular structure of compound **1**, along with the ^1^H-{^11^B} selective experiments and the observed broadening patterns, has facilitated the complete assignment of the proton resonances to their respective positions within the structure of **1**.

In Figure 4, it is observed that the ^1^H resonances assigned to the B1,3 *exo*-hydrogen atoms are grouped together between −0.51 and +1.30 ppm, forming a “low-frequency” cluster. On the other hand, the B4–H_exo_ signals are grouped between +1.83 and +4.81 ppm, forming a “high-frequency” cluster. Within this high-frequency group, the B4 *exo*-hydrogen resonance experiences significant deshielding when the metal atom is Nb, Re, Os, or Ir. This results in a large chemical shift difference for this particular resonance between [Rh(*η*^2^-B_3_H_8_)(H)_2_(PPh_3_)_2_] (**1**) and [Ir(*η*^2^-B_3_H_8_)(H)_2_(PPh_3_)_2_], as well as between [Ru(*η*^2^-B_3_H_8_)(CO)(PPh_3_)_2_H] and [Os(*η*^2^-B_3_H_8_)(CO)(PPh_3_)_2_H]. If we consider that the metal center is located antipodal to the B4 vertex through an axis connecting M2 and B4, this effect can be attributed to the change from a second-row transition metal center to a third-row transition metal center.

An interesting observation was made regarding the anomalous low proton shielding of *exo*-terminal protons that are positioned antipodal to third-row metal centers in twelve-vertex *closo*-metallaheteroborane systems. This phenomenon has been recognized as a diagnostic characteristic of this structural feature [24,25,26,27,28,29,30]. Similarly, we can utilize the strong deshielding of B4-H*_exo_* protons as a diagnostic indicator for the presence of third-row transition metal centers in four-vertex *arachno*-2-metallatetraboranes. This provides valuable insights into the structural composition of these compounds.

### 2.4. Fluxional Behavior

In order to investigate the chemical non-rigidity and fluxional behavior of compound **1**, a variable temperature (VT) NMR study was conducted in CD_2_Cl_2_. Figure 5 illustrates the changes observed in the ^1^H-{^11^B} NMR spectrum as the temperature was varied. The proton signals corresponding to B4-H*_endo_*, B1,3-H*_exo_*, and B4-H*_bridging_*-B1/B3 hydrogen atoms gradually broaden and eventually disappear, indicating an intramolecular proton exchange process in compound **1**. Notably, this process does not involve the B4-H*_exo_* and Rh-H-B1/B3 bridging hydrogen atoms nor the Rh-H hydride ligands.

The coalescence temperature, determined as the point at which the ^1^H signals merge, was estimated to be 300 K. Using this information, the activation energy (ΔG ^‡^) for the asymmetric population system was calculated to be 10 kcal/mol (see Appendix A for the analysis) [31].

To investigate the possible exchange of B4–H*_exo_*, the Rh–H*_briding_*–B1,3 and the Rh–H hydrogen atoms at higher temperatures, NMR spectra of compound **1** were measured at +67 °C in deuterated 1,1,2,2-tetrachloroethane. The ^1^H–{^11^B} spectrum revealed the formation of a new hydridorhodatetraborane, exhibiting proton resonances at *δ*_H_ −6.78 and −11.67 ppm, which were assigned to Rh-H-B and Rh-H hydrogen atoms, respectively (Appendix A).

A further increase in the temperature to +97 °C resulted in the decomposition of both compound **1** and the new hydridorhodaborane. The products of this decomposition included borane triphenylphosphine (Ph_3_P–BH_3_) and hydride-ligated complexes, as evidenced by the presence of several doublets in the ^31^P-{^1^H} spectrum (Appendix A). Additionally, the ^11^B NMR spectrum showed peaks between *δ*_B_ +2.5 and +10.0 ppm, which did not exhibit ^1^*J*(^11^B–^1^H) coupling, suggesting the formation of species containing O-B bonds (Appendix A).

The intramolecular hydrogen atom exchange observed in compound **1** shares similarities with the reported behavior of octahydridotriborate complexes such as [Mn(*η*^2^-B_3_H_7_Br)(CO)_4_] and [Ru(*η*^2^-B_3_H_8_)(CO)(H)(PPh_3_)_2_] [32], where the M2–H–B1/B3 bridging atoms also remain static. In these cases, the fluxional process occurs with an activation energy, Δ*G*^‡^, of approximately 12.2 kcal/mol at +23 °C for the manganesaborane. Interestingly, analogous compounds of third-row transition elements, such as [Os(*η*^2^-B_3_H_8_)(CO)(H)(PPh_3_)_2_] and [Ir(*η*^2^-B_3_H_8_) (H)_2_(PPh_3_)_2_], which are CO-ligated ruthenaborane and compound **1** analogues, respectively, do not exhibit fluxional behavior.

Several mechanisms have been proposed to explain hydrogen exchange in four-vertex *arachno*-2-metallatetraboranes. In the case of covalent metal-octahydridotriborate Be(B_3_H_8_)_2_, for instance, a rearrangement involving a Be-to-B_3_H_8_ bond change from *η*^2^ to *η*^1^, facilitated by Be–H–B bonds, followed by hydrogen atom exchange around the two BH_3_ units of the Be–{*η*^1^-B_3_H_8_} fragment, has been suggested. This mechanism ultimately leads to complete proton and boron exchange at high temperatures [33]. However, this mechanism cannot be applied to explain the observed exchange in compound **1**, as the Rh–H–B hydrogen atoms do not participate in the dynamic process.

Similar fluxional behavior has been observed in L_n_CuB_3_H_8_ species, where low-energy exchange of hydrogen and boron atoms occurs [34]. This behavior is reminiscent of the “free” B_3_H_8_^−^ anion, for which the energy barrier for complete scrambling of hydrogen and boron atoms has been calculated as 5.2 kcal/mol [35,36]. The fluxional process in copper-octahydridotriborate complexes involves a *pseudo*-rotatory motion of the {L_n_Cu} fragment around the {B_3_H_8_} ligand, supported by Cu–H–B bonds of different hapticity. Additionally, (CH_3_)_2_GaB_3_H_8_ and (CH_3_)_2_AlB_3_H_8_ have been found to exhibit fluxional behavior in solution, and the mechanism explaining the intramolecular exchange of hydrogen and boron atoms also involves metal-to-octahydrotriborane hapticity [37].

The fluxional process observed in the complex [Mn(*η*^2^-B_3_H_7_Br)(CO)_4_], where the Mn–H–B hydrogen atoms are not involved in the exchange, was proposed to involve a rotation around the B4–Br_exo_ bond coupled with rotation about either B–H bond in the metal-boron bridge. Similarly, in the case of compound **1**, we can propose a concerted rotation of the B4–H_endo_, B1,3–H_exo_ and B4–H–B1,3 bridging hydrogen atoms around the B4–H_exo_ bond as a mechanism to explain the observed fluxional exchange (Figure 2). This rotational motion would allow for the dynamic rearrangement of hydrogen atoms without involving the Rh–H–B or Rh–H–B1,3 bonds.

### 2.5. Reactions of 1 with Lewis Bases

We conducted preliminary and exploratory studies on the reactivity of *arachno*-2-rhodatetraborane (**1**) with pyridine (py) and dimethylphenylphosphine (PMe_2_Ph). The reactions were performed on a small scale in NMR tubes, and the results presented and discussed in this section should be considered as initial findings.

The ^31^P-{^1^H} NMR spectrum of the reaction mixture, obtained by adding pyridine to a CD_2_Cl_2_ solution of **1** in a 5 mm NMR tube at 233 K, reveals a doublet at *δ*(^31^P) +47.2 ppm, along with the resonances of the starting rhodatetraborane (Appendix A). In the ^1^H-{^31^P} NMR spectrum at 223 K, two new signals appear at *δ*(**^1^**H) −16.89 and −17.96, with a 1:1 relative intensity ratio, exhibiting the patterns of a *pseudo*-triplet and a doublet of doublets (dd), respectively (Appendix A). In the ^1^H-{^11^B} NMR spectrum, the apparent triplet transforms into an apparent quintet, and the dd becomes a triplet of doublets (Appendix A). The two-dimensional ^1^H–^31^P-HMBC spectrum reveals clear cross peaks between the ^31^P doublet and the two hydride signals, and the ^1^H-^1^H correlation further confirms the coupling between both hydrides (Appendix A).

In the ^11^B NMR spectrum, a broad peak of low intensity is observed at *δ*(^11^B) +19.3 ppm, accompanied by smaller intensity peaks between +2 and –5 ppm. The highest intensity signals correspond to a quartet at *δ*(^11^B) −12.1 p.p.m. and a multiplet at −37.8 ppm. The latter signal transforms into a doublet under ^1^H decoupling (Appendix A). The main ^11^B resonances can be confidently assigned to the pyridine and phosphine adducts, py-BH_3_ and PPh_3_-BH_3_. However, the assignment of the lower intensity triplet at −9.4 ppm (close to the quartet of the pyridine borane) remains uncertain.

Based on the observed NMR data, it is proposed that the reaction of compound **1** with pyridine results in the formation of borane adducts and a new cationic rhodium(III) complex, [Rh(H)_2_(PPh_3_)_2_(py)_2_]^+^ (compound **2**), which exhibits an octahedral structure (Figure 3). However, the formation of anionic species, in particular the borate anions, is not clearly observed in the NMR spectra. The low-intensity signals observed in the ^11^B NMR spectrum, some of which do not show ^1^*J*(^1^H-^11^B) coupling and others that appear as triplets, could potentially correspond to borate anions. Further characterization is required to determine the exact nature of the anionic species formed in the reaction with pyridine.

Upon addition of PMe_2_Ph to a CD_2_Cl_2_ solution of compound **1**, the ^31^P-{^1^H} NMR spectrum shows the appearance of two new doublets of triplets at *δ*(^31^P) +0.3 and −10.6 ppm. The spectrum at 233 K also reveals signals corresponding to free PMe_2_Ph and PPh_3_ at *δ*(^31^P) −45.6 and −7.3 ppm, respectively (Appendix A). In the ^1^H-{^11^B} spectrum, a new hydride resonance is observed at *δ*(^1^H) −10.15 ppm, exhibiting the pattern of a doublet of *pseudo*-quartets. This hydride resonance appears as a simple doublet in the ^1^H–{^31^P} spectrum (Appendix A). These observations strongly suggest the formation of the octahedral rhodium(III) cationic complex [Rh(H)_2_(PMe_2_Ph)_4_]^+^, in which the hydride ligands occupy *cis* positions to each other (Figure 3).

The ^11^B NMR spectrum exhibits a septet at *δ*(^11^B) −30.5 ppm, which can be assigned to the free octadecahydridoborate anion, B_3_H_8_^−^. There is also a multiplet at −37.7 ppm, which becomes a doublet upon ^1^H decoupling, corresponding to PhMe_2_P–BH_3_. Additionally, the spectrum shows signals of low intensity at −40.6, −44.7 and −45.4 ppm, as well as a broad peak of higher intensity at 16.3 ppm (Appendix A). These signals may correspond to uncharacterized metallaborane species present at low concentrations.

Overall, the data described for the reaction between **1** and PMe_2_Ph strongly suggest the formation of the salt [Rh(H)_2_(PMe_2_Ph)_4_][B_3_H_8_] (**3**).

## 3. Conclusions

The reaction in ethanol of Cs[B_3_H_8_] with the Wilkinson’s catalyst provides a convenient method for the preparation of the *arachno*-2-rhodatetraborane, **1**. This reaction involves the oxidative addition of two hydrogen atoms to the rhodium(I) center to form a {Rh(III)(H)_2_(PPh_3_)_2_}^+^ cationic fragment that binds the [B_3_H_8_]^−^ anionic ligand. The origin of the two additional hydrogen atoms is unclear and we can envision that some of the octahydridotriborate anion could donate them. Alternatively, the presence of ethanol could potentially act as a hydrogen transfer agent, facilitating the addition of hydrogen atoms to the rhodium center.

In the crystal structure, the sextuple phenyl embrace is an important driving force leading to the formation of ribbons in the lattice.

The fluxional behavior observed in compound 1 is similar to the non-rigid behavior found in other *arachno*-2-metallatetraboranes; based on the literature, we have proposed a probable mechanism of H atom exchange that involves the *endo*-H5 hydrogen atom, the *exo*-H1 and *exo*-H3 as well as the B1-H1,4-B4 and B3-H3,4-B4 bridging hydrogen atoms. It has been found that **1** is thermally unstable, decomposing at temperatures between +67 and +97 °C; this behavior suggests that the rhodatetraborane may exhibit a rich reaction chemistry *versus* different reagents.

We have explored this hypothesis in reactions of **1** with the Lewis bases, dimethylphenylphosphine, PMe_2_Ph and pyridine. In these reactions, we have found that the {*η*^2^-B_3_H_8_} anionic ligand is labile, and it is cleaved by PMe_2_Ph to form the salt [Rh(H)_2_(PMe_2_Ph)_4_][B_3_H_8_] (**3**). Alternatively, the reaction with pyridine demonstrates that dismantling of the *η*^2^-B_3_H_8_^−^ ligand can also lead to the formation of pyridine and triphenylphosphine adducts, L–BH_3_, and to cationic complexes such as [Rh(H)_2_(PPh_3_)_2_(py)_2_]^+^ (**2**).

The observed fluxional behavior and thermal instability highlight the versatility and potential reactivity of the rhodatetraborane compound **1**, making it an interesting candidate for further exploration in various chemical reactions and applications.

## 4. Materials and Methods

### 4.1. General

Reactions were carried out under an argon atmosphere using standard Schlenk line techniques. Solvents were obtained from a Solvent Purification System from Innovative Technology Inc. NaB₃H₈·3(C_4_H_8_O_2_) was purchased from Katchem spol. s r. o., and used as received. The deuterated solvent CD_2_Cl_2_ was deaerated, following freeze–pump–thaw methods, and dried over 3 Å molecular sieves.

Infrared spectra were recorded on a Perkin-Elmer 100 spectrometer, using a Universal ATR Sampling Accessory. Solution NMR spectra were recorded on Bruker Avance AV 300-MHz, AV 400-MHz and AV 500-MHz spectrometers, using ^11^B, ^11^B-{^1^H}, ^1^H, ^1^H-{^11^B}, ^1^H-{^11^B(selective)}, ^1^H-^31^P-HMBC and ^1^H-^1^H-COSY techniques. The ^1^H NMR chemical shifts were measured relative to the partially deuterated solvent peaks but are reported in ppm relative to tetramethylsilane. ^11^B chemical shifts are quoted relative to [BF_3_·OEt_2_].

### 4.2. Crystal Structure Determination

X-ray diffraction data were collected on an APEX DUO Bruker diffractometer, using graphite-monochromated Mo Kα radiation (λ = 0.71073 Å). Diffracted intensities were integrated [38] and corrected for absorption effects using the multi-scan method [39,40]. Both programs are included in the APEX4 package. All the structures were solved by direct methods with SHELXS [41] and refined by full-matrix least squares on F2 with SHELXL [42]. Hydrogen atoms were located from difference Fourier maps and refined isotropically.

Single crystals of **1** suitable for X-ray analysis were grown in a 5 mm NMR tube in a fridge at 4 °C by slow diffusion of hexane into a CH_2_Cl_2_ solution of the salt.

Structural data for [Rh(*η*^2^-B_3_H_8_)(H)_2_(PPh_3_)_2_]·2CH_2_Cl_2_ (**1**·2CH_2_Cl_2_, 100 K): Mr = 839.81, colorless prism, triclinic *P*−1, a = 12.3834(12) Å, b = 12.9413(12) Å, c = 13.9452(13) Å, α = 76.211(2)°, β = 85.014(2)°, γ = 66.7050(10)°, V = 1993.4(3) Å^3^, Z = 2, T = 100(2) K, Dcalcd = 1.399 g cm^−3^, µ = 0.803 mm^−1^, absorption correction factors min. 0.824 max. 0.924. 32,874 reflections, 8938 unique (*R*_int_ = 0.0659), 6581 observed, *R*_1_ = 0.0558 [*I* > 2σ(*I*)], wR_2_(*F^2^*) = 0.1562 (all data), GOF = 1.060. CCDC 2281251.

### 4.3. Mass Spectrometry

The mass spectrum for compound **1** was measured on a Thermo-Finnigan LCQ-Fleet Ion Trap instrument using electrospray ionization (ESI) with samples dissolved in acetonitrile (approximately 100 ng mL^−1^) and introduced to the ion source by infusion at a rate of 6 μL min^−1^: source voltage 3.2 kV, tube lens voltage −90.7 V, capillary voltage −32.0 V, capillary temperature 360 °C, drying gas flow 7 L min^−1^.

### 4.4. Computational Details

The calculations were performed using the Gaussian 09 package [43]. The structure of the model molecule, [Rh(B_3_H_8_)(H)_2_(PH_3_)_2_], was initially optimized using standard methods with the B3LYP/6-31+G(d) methodology and basis sets. The final optimization, including frequency analyses to confirm the true minima, together with GIAO nuclear-shielding calculations, was performed using B3LYP methodology with the 6-31++G(d) basis-set. GIAO nuclear shielding calculations were performed on the final optimized geometry, and computed ^11^B shielding values were related to chemical shifts by comparison with the computed value for B_2_H_6_, which was taken to be *δ*(^11^B) +16.6 ppm relative to the BF_3_(OEt_2_) = 0.0 ppm standard.

### 4.5. Preparation of [Rh(η^2^-B_3_H_8_)(H)_2_(PPh_3_)_2_] (***1***)

*Method A*: White powdery CsB₃H₈ [44] (0.0808 g, 0.470 mmol) was added to 10 mL of ethanol in a Schlenk tube (the ethanol was previously degassed with argon for five minutes). The tube was slightly heated to facilitate the formation of a solution, upon which [RhCl(PPh₃)₃] (0.4311 g; 0.470 mmol) was added to form a red-orange suspension. The reaction mixture was stirred at room temperature for three hours to give a yellow solid in suspension. The product was filtered through a frit, in air, to yield a yellow-mustard solid and an orange filtrate. The solid was collected in a Schlenk tube, dissolved in dichloromethane and filtered, under argon, through a silica gel layer. The resulting yellow solid was crystallized from CH_2_Cl_2_/Hexane (1:2). This final product was studied using NMR spectroscopy, demonstrating that its composition corresponded to the hydridrorhodatetraborane **1**. The total yield after drying under vacuum for several hours was 0.26 mg (0.387 mmol, 82.34%).

^1^H-{^11^B} (500 MHz, CD_2_Cl_2_, 223 K): *δ* + 7.66 to +7.32 ppm (m, aromatic signals, C_6_H_5_, 30H). IR (ATR): ν_max_/cm^−1^ 3054–2962 (w, C-H), 2508, 2452, 2378 (s, BH), 2451 (s, BH), 2418 (s, BH), 1585, 1568, 1480 (C=C aromatics). HR-MS(ESI): m/z calcd exact mass for C_36_H_40_B_3_P_2_Rh, [M]+, 670.1939; this anticipated parent ion is clearly absent. Instead, the spectrum exhibits high intensity peaks at 627.0867, 628.0884 and 629.0901 u, with an isotopic pattern that matches well that calculated for the ion [C_36_H_30_P_2_Rh]^+^, [M − (B_3_H_8_ + H_2_)]^+^. This ion corresponds to the {Rh(PPh_3_)_2_} fragment, demonstrating that the rhodatetraborane **1** undergoes facile cleavage upon ionization (Appendix A).

*Method B*: White powdery NaB₃H₈·3(C_4_H_8_O_2_) (0.1261 g, 0.385 mmol) was dissolved in 10 mL of dry ether, in a Schlenk tube, which was immersed in an isopropanol bath at −30 °C. Subsequently, the Wilkinson’s catalyst was added (0.3559 g, 0.385 mmol), under a flow of argon, to the sodium octahydridotriborate dioxane solution. The resulting brick-red suspension was stirred at room temperature under an atmosphere of argon. After one hour of stirring the temperature was increased to +5 °C and the color of the suspension became orange-red. The reaction was maintained for another three hours to give a brown-yellow suspension, immersed in the isopropanol bath at +10 °C. We allowed the solid to settle down and decanted the supernatant with a pipette, under a flow of argon. The decanted liquid was dried under vacuum to give an orange-yellow solid, whereas the sediment formed a beige solid, after drying. The NMR spectra of the former fraction (the decanted liquid) showed the presence of O=PPh_3_ and Ph_3_P–BH_3_, as major and minor components, respectively. The ether-insoluble product corresponded to the four-vertex rhodatetraborane. This method afforded 20 mg of **1** (8%).

### 4.6. Reactions of [Rh(η^2^-B_3_H_8_)(H)_2_(PPh_3_)_2_] (***1***) with Lewis Bases

*Reaction of [Rh(η^2^-B_3_H_8_)(H)_2_(PPh_3_) _2_] (***1***) with py*. 10.2 mg of 1 (1.50 × 10^−^^2^ mmol) treated with 1.20 μg of pyridine (1.50 × 10^−^^2^ mmol), in a Schlenk tube immersed in a bath of isopropanol at −30 °C. The resulting yellow solution was stirred, under an atmosphere of argon for 5 h; during this time, the temperature was raised to +10 °C. The reaction was stirred for another 20 min at room temperature. The solvent was evaporated under vacuum to give an orange solid, which was dissolved in deuterated dichloromethane and studied using NMR spectroscopy. ^31^P-{^1^H} (162 MHz, 233 K): *δ* +47.2 ppm [^1^*J*(^31^P-^103^Rh) = 118 Hz], together with the signal of O=PPh_3_. ^1^H-{^31^P} (400 MHz, 233 K): *δ* +8.59 (d, *J* = 5.0 Hz, *ortho*-NC_5_H_5_, 2H), +8.59 (d, *J* = 5.0 Hz, *o*-NC_5_H_5_, 2H), +8.34 (br. s, *o*-NC_5_H_5_, 2H), +7.95 (t., *p*-NC_5_H_5_, 2H), +7.95 (t., *p*-NC_5_H_5_, 2H), +6.54 (br. s, *m*-NC_5_H_5_, 2H), between +7.73 and +6.80 (m, C_6_H_5_-rings and NC_5_H_5_), −16.89 (t, ^1^*J*(^103^Rh-^1^H2,6) + ^2^*J*(^1^H2-^1^H2,3) = ^2^*J*(^1^H6-^1^H1,2) = 12.9 Hz, 2H) and -17.96 (dd, ^1^*J*(^103^Rh-^1^H2,6) = 23.4, ^2^*J*(^1^H2-^1^H2,3) = ^2^*J*(^1^H6-^1^H1,2) = 10.9 Hz, 2H). ^1^H-{^11^B} (400 MHz, 233 K): *δ* −16.89 (app. quintet, ^1^*J*(^103^Rh-^1^H2,6) + ^2^*J*(^31^P-^31^P) + ^2^*J*(^1^H2-^1^H2,3) = 12.9 Hz, 2H) and −17.96 (td, *J* = 25.5, 13.1 Hz, Rh-H2). *δ* ^11^B (400 MHz, 298 K): *δ* +18.8 (br.), +1.47 (s), −1.5 (t), −9.31 (t, 95 Hz), −12.0 (q, ^1^*J*(^11^B-^1^H) = 98 Hz, py-BH_3_), −37.8 (dq, 62 Hz, Ph_3_PBH_3_), −27.4 (t, 98 Hz).*In situ characterization of [Rh(H)_2_(PMe_2_Ph)_4_][B_3_H_8_] (***3***)*. 12.6 mg of 1 (1.88 × 10^−2^ mmol), dissolved in CD_2_Cl_2_, in a NMR tube, which was immersed in an isopropanol bath at −30 °C, and 2.6 mg (2.7 μL) of PMe_2_Ph (1.88 × 10^−^^2^ mmol) was added into the NMR tube, under a flow of argon. The reaction was studied using NMR spectroscopy, starting at 233 K and then heating the sample to room temperature. ^31^P-{**^1^**H} (162 MHz, 233 K): *δ* +26.7 ppm [s, O=PPh_3_], +19.7 (very br., PhMe_2_P-BH_3_), +0.3 [dt, ^1^*J*(^103^Rh–^31^P) = 97 Hz, ^2^*J*(^31^P1–^31^P2) = 24 Hz], −7.2 (s, PPh_3_), −10.6 p.pm [dt, ^1^*J*(^103^Rh–^31^P) = 86 Hz, ^2^*J*(^31^P1–^31^P2) = 24 Hz], together with the signals of 1 (Table 1). ^1^H-{^31^P} (400 MHz, 233 K): *δ* +7.84 to 6.94 (m, aromatics, C_6_H_5_), +1.56 (s, CH_3_), +1.46 (s, CH_3_), −10.25 (d, ^1^*J*(^103^Rh-^1^H) = 13.8 Hz. ^1^H-{^11^B} (400 MHz, 233 K): +1.2 (d, ^2^*J*(^31^P–^1^H), PhMe_2_-B*H*_3_), +0.23(s, B_3_H_8_^−^), −10.25 (d of *pseudo*-quintets, second order, ^2^*J*(^11^P-^1^H_trans_) = 147.9 Hz, ^1^*J*(^103^Rh-^1^H) = 14.0 Hz, ^2^*J*(^11^P-^1^H_cis_) = 17.2 Hz, 2H) and −17.96 (dd, ^1^*J*(^103^Rh-^1^H2,6) = 23.4, ^2^*J*(^1^H2-^1^H2,3) = ^2^*J*(^1^H6-^1^H1,2) = 10.9 Hz, 2H) ppm. ^11^B (400 MHz, 298 K): *δ* −16.4 (br. s), −30.5 (sept, B_3_H_8_^−^), −37.7 (quartet of d, ^1^*J*(^11^B-^31^P) = 59 Hz, ^1^*J*(^11^B-^1^H) = 102 Hz, PhMe_2_P–BH_3_), −45.2 (br. s).

## Data Availability

Not applicable.

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
