# Peer review of "The Synthesis, Characterization, and Fluxional Behavior of a Hydridorhodatetraborane"

_molecules, 2023, doi:10.3390/molecules28186462_

Round 1
Author Response
"Please see the attachment."

Reviewer 2 Report
The presented work is devoted to the preparation of coordination compounds of the [B3H8]- anion. Separately, it is worth noting the detailed analysis of structural characteristics of already known complexes of similar type and identification of regularities in the structure.
There are also several remarks to the work:
1. In the presented reaction scheme, rhodium changes its oxidation degree from +1 to +3. It is worth adding in the section of the discussion of the results a suggested mechanism (due to what oxidation occurs) or a reference to similar processes.
2. When comparing chemical shifts in NMR spectra of the obtained and described in the literature compounds, whether the parameters of spectra recording - solvent, temperature, concentration - were considered. It is worth mentioning it in the text.
3. The authors are requested to use prediction program with higher accuracy for calculation of peaks in mass spectra. If high-resolution mass spectra were obtained for the samples, convergence should be observed already in hundredths of a.u.m.
After corrections are made, the article could be published.
Author Response
"Please see the attachment."
